

# Regional-scale brine migration along vertical pathways due to CO₂ injection – Part 1: the participatory modeling approach

Dirk Scheer[1], Wilfried Konrad[1], Holger Class[2], Alexander Kissinger[2], Stefan Knopf[3], and Vera Noack[3]

[1]DIALOGIK, Lerchenstraße 22, 70176 Stuttgart, Germany
[2]Department of Hydromechanics and Modelling of Hydrosystems, University Stuttgart, Pfaffenwaldring 61, 70569 Stuttgart, Germany
[3]Bundesanstalt für Geowissenschaften und Rohstoffe (BGR), Stilleweg 2, 30655 Hannover, Germany

*Correspondence to:* Dirk Scheer (scheer@dialogik-expert.de)

**Abstract.** Saltwater intrusion into potential drinking water aquifers due to the injection of $CO_2$ into deep saline aquifers is one of the hazards associated with the geological storage of $CO_2$. Thus, in a site-specific risk assessment, models for predicting the fate of the displaced brine are required. From the very beginning, this research on brine migration, has been aimed at involving expert and stakeholder knowledge in simulating the impacts of injecting $CO_2$ into deep saline aquifers

by means of a participatory modeling process. The involvement exercise made use of two approaches. First, guideline-based interviews were carried out aimed at eliciting expert and stakeholder knowledge and assessments on geological structures and mechanisms affecting $CO_2$ induced brine migration. Second, a stakeholder workshop, including the world café format, was used to evoke evaluations and judgments on the modeling approach, on scenario selection, and on preliminary simulation results. The participatory modeling approach gained several results covering brine migration in general, the geological model

sketch, scenario development, and the review of the preliminary simulation results.

## 1 Introduction

Any effort in investigating and developing the Carbon Dioxide Capture and Storage technology (CCS) unavoidably touches the social and political spheres, and needs to take into account the broader societal debate. From the very beginning, this research on brine migration, has been aimed at involving expert and stakeholder knowledge in simulating the impacts of injecting $CO_2$

into deep saline aquifers. Therefore, this work is split into two papers (Part 1 and Part 2), where Part 1 deals with the concept of "Participatory Modeling" (PM) as a means to involve external experts and stakeholders in the modeling process and Part 2 deals with the technical findings relevant for modeling brine migration. The main objective of Part 1 is to introduce participatory modeling in a joint natural and social science approach as a means to involve potential stakeholders of $CO_2$ storage applications into the technical modeling process.

In essence, this study focuses on a comprehensive participatory stakeholder and modeling process, investigating scenarios and model approaches for regional-scale brine migration on the groundwater system due to $CO_2$ injection. The basis of this study is a realistic (but not real) virtual site derived from a geological model with geological structures as can be found in the North German Basin. After Knopf et al. (2010), the North German Basin is considered the most relevant region regarding





$CO_2$ storage capacity in Germany. The adopted geological model comprises layers from the deep saline injection horizon up to shallow freshwater aquifers. For the numerical simulations of brine migration, the model fully couples fluid flow in shallow freshwater aquifers with deep saline aquifers. Within this system, we investigated different scenarios that can lead to brine migration into shallow aquifers. A site-specific assessment of potential hazards, such as this one, would be necessary in the

early phases of a multi-stage site identification process.

Public acceptance and a profound understanding of risks, hazards, and benefits are key issues on the path to realization of such projects (Scheer and Konrad, 2014). Therefore, it is good practice to already involve stakeholders at an early stage during the site-identification process (Scheer et al., 2015). When applying PM, we incorporate, from the very beginning of the modeling process, stakeholder expertise and opinion making, helping to reflect the geological model setup and relevant

scenarios describing brine migration. As PM describes both a societal and technical approach, references will be made to both social scientists and modeling experts. When using the term 'modelers', we refer to the authors of this study who have a background in the field of geology and numerical modeling.

The concept of PM provides a framework for integrating external expertise into producing and deploying conceptual and computer-based models (Bots and Daalen (2008); Dreyer et al. (2015); Dreyer and Renn (2011); Röckmann et al. (2012)).

PM is a generic approach, open for different methods in order to facilitate early expert and stakeholder integration in science development. This integration of external expertise in geo-science development is currently still a rather exceptional case. We define PM as integrating experts and stakeholders into the production and/or usage phase of conceptual and computer-based models (Hare et al. (2003); Bots and Daalen (2008); Dreyer et al. (2015)). Hence, PM opens up the modeling process for external actors whose expertise lies outside the realm of simulation and modeling science. In that sense, PM is a generic term

for a large variety of experimenting with expert involvement in science development. PM comprises several approaches, such as "Group Model Building" focusing on strategy development in organizations (Richardson and Andersen, 1995), "Mediated Modeling" with its aim to generate consensus for environmental issues (van den Belt, 2004) or Companion Modeling for collective learning in the field of natural resource management (Simon and Etienne, 2010). Most research of PM application currently takes place in the management of natural resources, such as water, forestry, or land use (e.g., Refsgaard et al. (2005);

Antunes et al. (2006); Cockerill et al. (2006); Bogner et al. (2011a); Webler et al. (2011); Röckmann et al. (2012)). To our knowledge, no PM stimulated applications have so far been carried out in the field of CCS. Nevertheless, some research has been done on identifying how policymakers process and use simulation results in the field of geological $CO_2$ storage (Scheer (2013); Scheer (2015)). Applying PM research follows, in general, two objectives as highlighted in literature (Dreyer and Renn, 2011). The first objective is to come to a robust and an ideally consensual and jointly born recommendation for

policy and management. This shall be done via the integration of expert and stakeholder related knowledge into the modeling process in order to improve the model quality. The second objective aims at stimulating collective learning processes within the involved stakeholder group. The general idea of PM fits well into our own research. However, with our approach, we build on experiences with other stakeholder elicitation processes in the field of CCS. For instance, in 2011, research was carried out applying a combination of a traditional Delphi survey and a group Delphi method that focused on a broader range of topics

such as technological challenges, administrative and legal aspects, chances and risks, societal relevance and communication





issues (Wassermann et al., 2011). In addition, an expert elicitation study was undertaken to identify, assess, and rank potential $CO_2$ leakage scenarios in Heletz, Israel, to provide guidance to support the decision-making processes (Edlmann et al., 2015).

Conclusions drawn from this short literature review indicate a lack of both methods and case studies covering early expert involvement in science development. Research carried out at the science-policy interface often involves difficulties in under-

standing among stakeholders and decision-makers. As such, the transfer of scientific concepts to the practical application can benefit from an early-stage expert evaluation. To elaborate adequate methods and carry out a case study in the field of CCS has been the main motivation of the research presented in this paper.

The involvement exercise undertaken within the modeling of different brine migration scenarios made use of two approaches. As a starting point, guideline-based interviews carried out by the social scientists aimed at eliciting expert and stakeholder

knowledge and assessments on geological structures and mechanisms affecting $CO_2$ injection-induced brine migration. The second involvement approach consisted of a stakeholder workshop, including the world café format, which was carried out in order to evoke evaluations and judgments on the modeling approach, on the scenario selection, as well as on the preliminary simulation results. The paper is organized as follows: Section 2 elaborates on the materials and methods used by shortly summarizing the participatory concept and outlining the detailed involvement steps and formats. Results of eliciting and feeding

back expert information are provided in Section 3. Following this, Section 4 discusses the main results, while Section 5 will end with a short conclusion.

## 2   Concept, methods and materials

### 2.1   The concept: early participatory modeling stakeholder involvement

The modeling process comprised both the setup of a static geological model and the implementation of dynamic numerical

models used for investigating different brine migration scenarios as defined by the national "Carbon Dioxide Storage Law" (KSpG, 2012). The modeling concept provided the opportunity to involve stakeholders at a very early stage of science development. Thus, within this study we had the chance to include stakeholder opinion making and critique in the elaboration process of the geological model, the numerical model (i.e. the relevant physical processes), and the brine migration scenario design. As such, the focus of early stakeholder involvement in the modeling process comprised to:

i.  critically assess and, if necessary improve our proposed geological model,

   ii.  critically reflect and thus contribute on brine migration scenario development and,

   iii.  critically review and discuss preliminary numerical simulation results.

The participatory modeling concept covered two involvement methods: several expert interviews and one expert workshop. Both approaches were assigned at decisive time spots within the science management process. Figure 1 details the combination

of the scientific and participatory processes. The science development first started with elaborating the preliminary sketch of the geological model, which served as input for the interviews. The interviewees critically assessed the sketch and provided





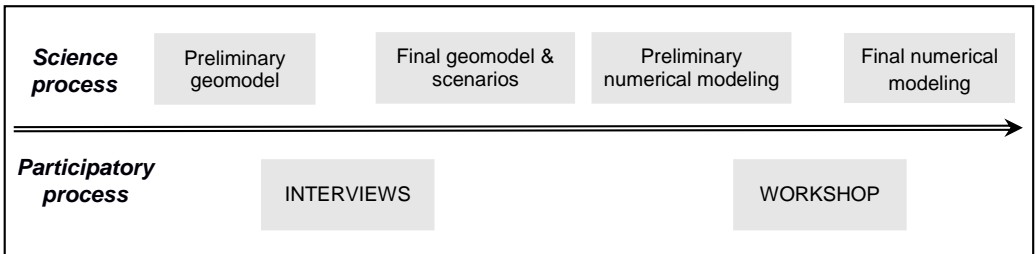

**Figure 1.** Detailing the science and participatory process concept.

expert insights on brine migration mechanisms and scenarios. The expert knowledge supported finalization of the geological model and the scenarios elaborated by the modelers, which were fed into the numerical modeling. The preliminary numerical modeling results were then critically discussed by stakeholders within the expert workshop. Subsequently, modeling results were finalized (Kissinger et al., 2017).

## 2.2 The methods: expert interview and expert workshops

*Expert interviews* are a permanent feature in the toolbox used in empirical social research (Mayring (1990); Bogner et al. (2011b)). Ten interviews were conducted between May and June 2013 by the social scientists with interviewees representing public authorities (5 interviews), business and industry (2 interviews), the scientific community (2 interviews), and a non-governmental organization (1 interview). The interviewees were provided with a questionnaire covering the following topics:

- Hazard assessment $CO_2$ injection: most important risks and hazards

- Brine migration mechanisms: pathways, physical processes, target variables

- Scenarios for brine migration: prioritization of brine migration pathways

- Geological model: review and recommendations

The interviews were conducted face-to-face and lasted on average 60 minutes. Key issues addressed by the questionnaire referred to parameters and processes influencing brine migration, and the specification and prioritization of brine migration scenarios. The social scientists provided interviewees with some detailed questions jointly compiled with the modelers along with the previously introduced model sketch shown in Figure 2 in order to get the stakeholders' critical feedback on their understanding of brine-related hazards, mechanisms and the plausibility of the principal geological model setup.





The *expert workshop* took place in Hannover in September 2014, where a total of 17 external participants and six project staff members gathered. External participants represented public authorities (8 participants), business and industry (2 participants), the scientific community (4 participants), and experts from non-governmental organizations (3 participants). Within the first session, modelers presented both the geological model as well as preliminary simulation results. During the second part of

the workshop, a world café deliberation was carried out. The "World Café" is a structured conversational process, which aims to facilitate open and intimate discussion. The idea behind a World Café is to provide access to the collective intelligence or collective wisdom among participants. Participants move between a series of tables where they continue the discussion in response to a set of questions, which are predetermined and focused on the specific goals of each World Café (Brown and Issacs (2005); Steier et al. (2015)). For that purpose, the participants in our workshop were divided into several small groups

seated around tables discussing predefined core questions. After 20 minutes, we recombined the groups in the way that each member of a group moved to a different table. Only one person, the host, remained at the table and informed the new group about what had happened in the previous discussions. This procedure was repeated three times.

Recruitment of experts for both the interviews and the workshop followed several selection rules: expertise in dealing with the topic of geoscience, (and/or) carbon dioxide capture and storage, (and/or) modeling; representing (either/or) the area of

public authorities, business and industry, scientific community, and the civil society; and have longstanding experience and/or hold a senior professional position.

### 2.3    Material for stakeholders: the geological model (sketch) and preliminary simulation results

Data derived from 3D geological models were used by the modelers for the construction of a sketch for the guideline-based interviews (Figure 2). The sketch already includes technical and geological features, which may provide pathways for brine,

such as abandoned wells, fault zones and hydrogeological windows in the Rupelian clay barrier.

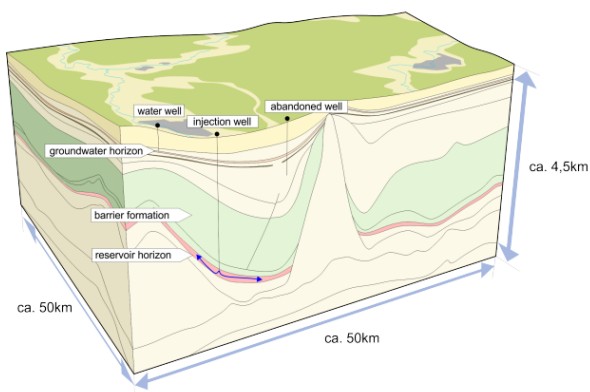

**Figure 2.** Sketch of the geological model used for the interviews (graphical realization: Jens Rätz).

The geological model presented at the stakeholder workshop did not consider a real site, but was based on a real structural configuration derived from the German North Sea. The model comprised layers from the deep saline injection horizon up



to shallow freshwater aquifers. The region belongs to the North German Basin and was affected by salt mobilization during different geological time periods. This mobilization also affected the geometry of the overburden. The result is an elongated anticlinal structure, which is meant to act as a structural trap for the injected $CO_2$. One of the key features of the geological model is a rising salt wall, which pierces through all layers up to the shallow freshwater aquifers. The geological model includes

two important barrier layers: the Upper Buntsandstein barrier, and the Rupelian clay barrier. The Upper Buntsandstein barrier is the first barrier above the injection horizon and prevents the injected $CO_2$ from migrating out of the injection horizon. The Rupelian clay barrier separates shallow freshwater aquifers from deep saline aquifers. We modified this hydraulic barrier to be penetrated by the uplifted Cretaceous sediments on top of the anticlinal structure (so-called hydrogeological windows). Making a conservative assumption, we assumed a permeable vertical pathway along the whole flank of the salt wall, which we refer

to as a fault zone. This fault zone is a permeable connection between the injection horizon and the shallow aquifers above the Rupelian clay barrier. The main reason for assuming this permeable fault zone along the salt wall, was a statement by LBEG (2012): "the contact zone between salt domes and the $CO_2$ -sequestration horizon is assumed to be a zone of weakness, similar to geological faults".

Based on the geological model, numerical simulations were carried out for different scenarios where the lateral boundary

conditions (no-flow boundary, constant-pressure boundary), the Upper Buntsandstein barrier permeability, and the fault zone permeability were varied. The numerical model at that stage did not consider the transport of salt nor its effect on the density and the viscosity of brine. Further, simplified models were presented and different model simplifications were compared, such as injecting brine instead of $CO_2$ and using an analytical model to calculate leakage through the fault zone. The target variables considered were the evolution of leakage rates over the fault zone and the hydrogeological windows during the injection.

Further, the spatial distribution of flow rates per unit area at the end of the injection period at the bottom of the shallow aquifers was considered. The main conclusions drawn from these preliminary results were that the boundary conditions of the system and the Upper Buntsandstein barrier permeability have a strong influence on the amount, and the location, of injection-induced leakage.

## 3  Results

The participatory modeling approach yielded several results covering brine migration in general, the geological model sketch, scenario development, and the review of the preliminary simulation results. In the following, we will expose the main results from the interviews and the workshop.

### 3.1  Brine migration: general issues

The interviews revealed several decisive issues tackling brine migration in general. A first result from the interviews relates

to the conceptualization of 'damage' in case saltwater would reach drinking water aquifers. Some stakeholders favored what we call an 'absolute' understanding, meaning that damage occurs as soon as any saltwater, regardless of the volume, intrudes drinking water aquifers. This group of stakeholders holds the opinion that any intrusion of brine must be considered a damage,





which implies an understanding of a zero-risk tolerance. Others hold the opinion that the salinization of groundwater needs to be considered in 'relative' terms. For the latter experts, damage is not a question of whether or not brine comes into contact with groundwater, but is rather defined as an event where specific threshold values are exceeded, in this case the volume of the intruded brine matters. In order to allow judgments of risk, a detailed assessment of the brine quantity, its salinity, and

probabilities of occurrence need to be performed. This issue remained largely unsolved during the interviews and hints to differing concepts, perceptions or interests that may frame the interviewees' risk-related thinking.

Concerning potential brine migration paths, the interviewed stakeholders unanimously made a clear-cut distinction between man-made and geology-induced hazards. The former comprises facilities such as old and new boreholes or drinking-water wells while geology-induced hazards refer to cracks and faults, salt diapirism/doming, thin and non-continuous seals or a

non-continuous Rupelian clay barrier. The distinction between potential migration paths caused by technical installations or geological structures was accompanied by a distinct hazard prioritization. All participants agreed in estimating geology-based hazards as far more relevant compared to man-made hazards. In general, interviewees argued that man-made hazards, such as a faulty drill hole, are much easier to cope with technologically and allow only relatively small quantities of brine to migrate. The main reason to assume that man-made hazards are less relevant is due to the perception that only very small brine volumes

are able to migrate through improperly plugged abandoned wells.

As for the main hazards of $CO_2$ injection with regard to brine displacement, interviewees stated consistently that vertical brine migration, salinization of groundwater, increase of pressure, and uplifting typify the most relevant hazards. However, in line with the differing understanding of risks and damages in general, interviewees stated a diverse set of reference points. One statement argued, for instance, that vertical brine migration is not a general hazard, but only related to specific sites. Another

statement linked brine migration issues more to the social world, arguing that it is a juridical, contamination, and data collection problem. Other statements referred to issues as what value should be protected and mentioned several subjects of protection such as the wildlife, people, water (drinking water, healing water, mineral water).

Considering target variables, the interviews brought together a great variety of target variables to be considered. First, interviewees stated that there is, in general, a need to determine what exactly an extent of damage is and to agree on relevant

target variables. However, the relevant target variables varied among interviewees. In sum, interviewees mentioned the use of variables such as salt concentration, several different types of ions, water quality indicators, and chlorine content, or the use of total dissolved solids (TDS) as an aggregate indicator, the electrical conductivity as a sum parameter, and the pressure variance.

## 3.2   Geological model issues

Interviewees were provided with a sketch of the intended geological model as elaborated by the modelers (Figure 2). The ge-

ological model sketch intended to initiate and stimulate discussions and reflections concerning model specifications thereafter implemented by the modelers. Interviewees, in addition, had the chance to draft some explanations and further illustrations on the paper sheet. Figure 3 shows an example of a model sketch commented on by an expert.

The geological model sketch and the interview guideline stimulated the discussion of several issues. First, on a generic level, interviewees noted the model sketch is far too simple, and greatly underestimates real-world complexity – though experts





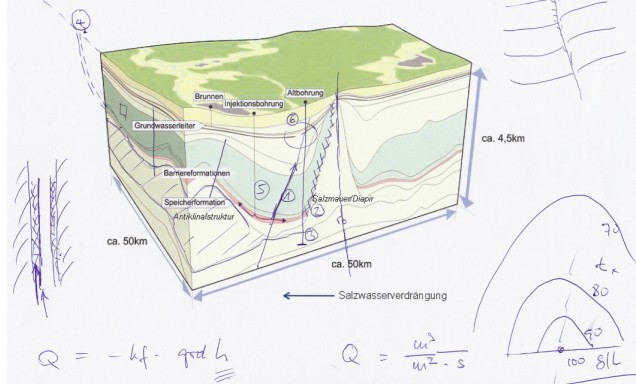

**Figure 3.** Example of commented geological model sketch by interviewee.

conceded that a model sketch at the corresponding research stage, i.e. the site-exploration stage, needs to be simple. Second, issues of model boundaries were raised. The modeling results are very sensitive to the specification of model boundaries. Thus, a critical reflection on the type and determination of model boundaries is essential. The sketch by itself was not specified with the type of model boundaries, leaving it open whether the aquifers are closed or open. Third, the issue of old boreholes was

discussed. As shown in the sketch, it contained just one abandoned well; it remains an open question whether this is enough as stated by an interviewee. In addition, the indicated water well in close proximity to the non-continuous Rupelian clay barrier was seen critical. Fourth, several geological issues were raised. Interviewees mentioned the Rupelian clay barrier might serve as a migration path depending on the level of pressure. The salt diapir in the sketch remained unclear concerning its three dimensional shape (wall, tapered, cylindrical, mushroom-like). Depending on the considered shape of the diapir, $CO_2$ is either

able or unable to spread and flow. Fifth, fault zones shown in the sketch were estimated far too little and in wrong places, and fractures and the geometry of fault damage zones were not specified. Concerning barriers, statements mentioned, that there is a need of two barrier formations according to the German CCS law (KSpG, 2012). Finally, interviewees recommended changing the location of the injection point. Recommendations for better locations include injection below the Zechstein salt (which is the bottom layer of the sketch) or injecting directly into the inflection point of the anticlinal structure – as opposed to injection

at the flank of the anticlinal structure, as suggested by the sketch.

### 3.3   Scenario issues

Numerical simulation of brine migration along vertical pathways was intended to conceptually run and compare different scenario settings varying with parameter values and/or initial boundary conditions. Interviews served to discuss and provide relevant issues for scenario design and development. As a result, interviews elicited a broad range of key elements for scenario

building. Table 1 depicts these elements together with stakeholders' suggestions on how to integrate them into brine migration scenario modeling. Social scientists fed back several stakeholder suggestions on the geological model, scenario design and alignment, and relevant geo-physical and geo-mechanical processes to the modelers. Modelers relied on these expert sugges-



**Table 1.** Key elements for scenario building from the interviews

| Element | Stakeholder suggestions |
|---|---|
| Boundary conditions | Consider different boundary conditions as they have a considerable impact on brine displacement and pressure increase mechanisms |
| Geological structure | Use different geological structures since brine displacement and pressure increases are highly dependent on the geological structure |
| Space dimensions | Investigate scenarios with different spatial dimensions (e.g. a large-scale scenario with 100 km) |
| Man-made migration paths | Integrate drill holes in order to validate expected impacts such as low displacement quantities and minor increase in pressure |
| Variable layer permeabilities | Vary permeabilities of important layers |
| Injection points and volumes | Consider different injection points and volumes |
| Pressure management | Simulate different volumes of brine production |
| Grid discretization | Work with detailed discretisation of geological weak points vs. rough discretisation of huge spatial structures |

tions and developed four different scenarios for running brine migration simulations. For details and results on the scenarios see Part 2 of the study (Kissinger et al., 2017).

## 3.4 Numerical simulation

Within the workshop, the principal conceptual design and first simulation results were presented to participants by the project team. Subsequently, we used a world café format to interactively discuss issues of the simulation concept. The world café group discussions centered on two sets of questions covering the spatial dimension of the model, the migration pathways at the flank of the salt diapir, and the conceptual approach of using a realistic but not site-specific geological model. An open discussion finally focused on first simulations results. In the following, we will present key findings of the stakeholder feedback from the workshop.

*First set of questions: "Basic assumptions of the geological model are the spatial dimension of 58 x 39 km and a permeable fault zone along the salt wall. How do you evaluate these assumptions? Is the spatial dimension sufficient to investigate pressure effects in the far-field of the $CO_2$ injection? Is brine migrating along a salt wall up to the top of the salt diapir realistic?"*

Stakeholder comments differed depending on whether one or multiple injection points were considered. In the case of injecting $CO_2$ at various sites, participants unanimously agreed, that due to pressure interference a wider space must be investigated. Contrasting opinions were raised for modeling just a single injection point. One group held the opinion that the assumed space size of 58 x 39 km is sufficient. They argued that the brine primarily follows vertical migration pathways. Other participants



challenged this argument by referring to studies that demonstrate a rise in pressure even in distances of 100 and more kilometers. According to this judgment, researchers should use models with spatial parameters of adequate scale in order to create reliable scenario findings. The discussion regarding brine migration pathways at the flanks of salt diapirs brought out contrary results with both opinions affirming and denying pathway probabilities. For some stakeholders, the existence of permeable

pathways along flanks of salt diapirs seems probable. Others were convinced that this is not a realistic assumption, and thus found it implausible to model leakage at the salt diapir. If permeable pathways along the salt wall exists, salt does not dissolve at the wall since water in contact with the diapir is already saturated. These contrasting views on the migration along the salt-diapir led to a final request that a comparative study be performed, varying the permeability parameters along the salt wall from low to high.

> *Second set of questions: "We consider a realistic, but not site-specific model. Is this, in your opinion, an appropriate approach for gaining general insights into brine migration with scenario modeling?"*

The majority of the stakeholders endorsed the modeling approach and confirmed that generic findings can be drawn from a realistic, but not site-specific, model. Key aspects in terms of processes, methods and structures are covered serving the model to be used for improving the understanding of fundamental issues, even before an exploration drilling takes place. Of course, stakeholders were aware that working with a realistic model does not substitute a site-specific analysis. However, this insight

was the starting point for a minority of participants, stressing that only geological on-site investigation would be able to deliver reliable findings.

*Final workshop discussion:* The final session of the workshop presented group work results and openly discussed the findings in full plenum. Here, stakeholders made the following additional comments on the preliminary simulation results:

- The injection of brine into a brine-filled storage horizon instead of $CO_2$ was considered to be a valid assumption

- The assessment of dynamic effects in the groundwater system during the injection of $CO_2$ is a valuable contribution for understanding pressure conditions and fluid migration processes in complex geological systems

- The stakeholders found it useful to identify the zones where highest local flow rates occur, if the effects of fluid and rock compressibility on the storage capacity of the system is exhausted

- The simulations should include the variable-density flow of brine

- Groundwater recharge as a boundary condition for the shallow aquifers should be considered

- Overlapping pressures from multiple injection sites should be considered

## 3.5   Feedback to and revisions of modelers

Social scientist gathered elicited expert knowledge and expertise and fed back recommendations to the modelers. The modelers were then required to review each statement and balance whether or not to revise their research. Modelers categorized





the stakeholder input according to four major categories: (A) Stakeholder issues, which were already considered within the preliminary simulation results. (B) Newly implemented issues after stakeholder workshop which were already planned. (C) Stakeholder issues that were initially not covered but during the participatory process are now seen as relevant by the modelers. (D) Stakeholder issues that were not realized, either because they were beyond the scope of the project or deemed less relevant.

5   Table 2 provides an overview on stakeholder input and issues as they were implemented or not in the research.





**Table 2.** Overview on implementation and revision of stakeholder input[1]

| Stakeholder input | Revision | Rationale |
|---|---|---|
| *Brine migration: general issues* | | |
| absolute vs. relative damage | A | zero impact is deemed impossible by modelers; results should be interpreted relative to salinization prior to injection |
| man-made vs. geology hazards | D | man-made hazards were considered much less important than geological hazards by the stakeholders |
| *Geological model issues* | | |
| model simplicity | A | modelers decided that pathways representative for the NGB (permeable fault zone at salt wall flank and hydrogeological windows in the Rupelian clay barrier) should be considered. It was also decided against including more pathways (e.g. leaky wells, more fault zones) as this would make the showcase too complicated for PM |
| model boundaries | C | domain extension (100 km) resulting in infinite aquifer-like conditions |
| Rupelian clay barrier | A | considered as second important barrier layer with discontinuities at hydrogeological windows |
| fault zones and fractures | A | permeable fault zone at the salt wall directly connecting injection horizon with shallow aquifers is considered |
| injection point | D | variable position of the injection within the anticlinal structure was not considered because it is deemed to be of minor relevance for large-scale brine migration by the modelers |
| *Scenario issues* | | |
| boundary conditions | A & C | scenario with variation of lateral boundary conditions (infinite aquifer, no flow and constant pressure) |
| space dimensions | A & C | the lateral extension of the model domain up to 100 km to obtain more realistic lateral boundary conditions (infinite aquifer) |
| variable layer permeabilities | A | scenario where permeability of important Upper Buntsandstein barrier is varied |
| Injection points and volumes | D | variable injection volumes/rates were not considered as brine migration rates; could be inter- or extrapolated (superposition) from the results |
| pressure management | D | beyond scope of the study |
| grid discretization | D | no refinement near geological weak points to maintain computational feasibility; comparison to analytical solution with similar discretization for simplified geological model yielded acceptable agreement |
| | | *continued on next page* |





| continued from previous page | | |
|---|---|---|
| **Stakeholder input** | **Revision** | **Rationale** |
| *Numerical simulation issues* | | |
| spatial dimension | C | lateral extension of the model domain up to 100 km to obtain more realistic lateral boundary conditions (infinite aquifer) |
| permeable salt wall flank (fault zone) | C | variable parametrization (permeability) of the fault zone along the salt wall; investigation of sensitivity of leakage depending on fault zone permeability is performed |
| brine injection | A | brine is injected at a volume-equivalent rate to the $CO_2$ injection rate |
| Pressure evolution | A | consideration of compressibility of solid and fluid phases, infinite aquifer boundary conditions |
| identification of areas prone to salinization | A+B | spatial distribution of flow rates per unit area and salt concentration increases |
| variable-density flow | B | density and viscosity are a function of the salt concentration |
| groundwater recharge | C | groundwater recharge for the top aquifers to establish more realistic flow conditions in the shallow formations |
| multiple injection sites with overlapping pressure | D | beyond scope; would require a basin scale model of the North German Basin which is not available yet |

## 4 Discussion

This study comprised a joint natural- and social-science research approach with the aim of involving stakeholders in the scenario development and modeling process at an early stage. The innovative design brought new insights both in the field of
5   natural science-based CCS research related to the hazard of brine migration, as well as in the field of social science-based inter- and transdisciplinary research areas. Hence, results from both fields are strongly connected.

First, we will shortly summarize the main findings from the geological and numerical modeling exercise in order to allow the readers a joint perspective. The results are based on the revised geological and numerical model that was designed within the PM process. A more extensive discussion of these results is provided in Kissinger et al. (2017). The main findings can be
10   summarized as follows:

- Notable, in the sense of non-negligible, increases in salt concentration in the target aquifers are locally constrained to regions, where initially elevated concentrations are present prior to the injection, and where permeabilities are high

---

[1]Explanation:

A) Already considered during presentation of preliminary results at stakeholder workshop

B) Newly implemented after stakeholder workshop (already planned)

C) Newly implemented after stakeholder workshop (initially not planned)

D) Not implemented because out of scope or deemed less relevant by stakeholders or modelers





enough to support sufficient flow. Hence, the quality of the prediction of concentration changes strongly depends on how well the initial salt distribution is known.

- An inherent problem to modeling is the assignment of boundary conditions. Lateral and top boundary conditions strongly determine the amount of displaced brine into the target aquifers. Lateral Dirichlet boundary conditions at insufficient distance from the injection will lead to a strong underestimation of vertical flow. Setting the top boundary condition as open – as opposed to a closed boundary at the top – strongly increases the amount of fluid that is displaced into the target aquifers.

- The permeability of the Upper Bundsandstein plays a crucial role in determining the amount of diffuse leakage. Diffuse migration through the Upper Buntsandstein barrier can result in focused leakage in locations where the Rupelian clay barrier is discontinuous.

- Injecting an equivalent volume of brine instead of $CO_2$ is a conservative assumption, which leads to slightly increased brine flow into the shallow aquifers and a reduced pressure buildup in the injection horizon.

Second, we more extensively discuss the main findings from the participatory approach as presented within this paper. The most important tool used within this research has been running simulations in order to analyze brine migration scenarios – the process integrated from the very beginning stakeholder involvement. The joint approach intended to gain new insights on geological matters, and – from a methodological perspective – gain insights on potentials and constraints of participatory modeling in the field of geo-science, and for participatory approaches in general.

By the time this research began (2012), the public debate on the geological storage of $CO_2$ was already in decline, as it was clear that there would be no large-scale $CO_2$ sequestration projects in Germany in the near future due to fierce public opposition and an inadequate regulatory framework. This also reduced the motivation of stakeholders to get involved in the PM process. Despite these adverse conditions, our research was able to attract the attention of a more general audience through a newspaper article published in one of Germany's major newspapers (Schrader, 2014).

The decision to conduct research on brine migration for a virtual site instead of a specific site influenced the recruitment of the expert panel. The group of participants comprised experts in the field of CCS and geo-science modeling representing the scientific community, regulators and public authorities, business and industry associations, and non-governmental organizations. Actors and stakeholders from a local level, such as members of the affected public, members of municipal and local counties, or representatives from local environmental groups of citizen initiative groups were not considered part of the participatory modeling process. The decision in favor of this type of participant recruitment strategy was due to the research objective of providing solid scientific methods backed by external expert knowledge, and minimize the politicized bias within the deliberation process. The composition of participants at the expert workshop also helped to create a "productive atmosphere", at least in the opinion of the authors. This means that the discussion was focused around the methodology and the results presented at the workshop, without drifting off into other CCS-related topics, which were beyond the scope of this research. In this way, the modelers were able to profit from the discussion through helpful suggestions and critical remarks. If a more



general public were involved in the PM process, more effort would be required during the preparation and the presentation of the methodology, and the benefit of the process, in terms of helpful scientific suggestions, would have been much smaller.

An important question in participatory modeling is whether or not to involve external experts within the model construction process. In most cases, model construction involvement is very much constrained, as the model already exists. In our case, we had the chance to integrate experts in the geological model construction phase. However, to be more precise, the impact of participants on model construction was limited to comments on, and recommendations towards, a given basic geological model. First, modelers decided to use a virtual model characteristic for the North German Basin, which was fed with geological data from 3D models of a region in the southwestern German North Sea. The main reasons for not involving participants in the decision have been twofold. First, the North German Basin is the most important area for potential CCS storage and hence is in line with state of the art CCS research. Second, the rights to use specific geological data were held by a research partner, so the model could be easily used for carrying out the analysis. On the other side, the geological model construction had to be further modified and specified based on the given geological dataset in order to run the intended simulations on $CO_2$ injection and brine migration. That is where the participatory modeling exercise came into play. Stakeholders contributed with their expertise towards improving the proposed geological model, the brine migration scenarios, and the final numerical results. From that end, stakeholders had a notable impact on the final geological model, and the design of brine migration scenarios.

## 5  Conclusions

Involving external experts and stakeholders in the evaluation of brine migration models by means of participatory modeling techniques has proven to be a helpful and successful approach. It led to valuable recommendations for the modelers' research and has enabled knowledge transfer to stakeholders. The groundwork for this positive outcome is the interaction between those three actor groups crucial for the performance of PM processes, i.e. modelers, stakeholders, and social scientists. Openness to stakeholder input and a general willingness to adapt models, concepts, or findings in response to stakeholder evaluations are key requirements for modelers in PM processes. This cannot be taken for granted, since the modelers have detailed insights into the problem setting. Hence, in order to be accepted by the modelers, the participating stakeholders must consist of experts, decision-makers, or affected people well known for their expertise in the respective field. Although stakeholders are required to be experts themselves, they need to agree with predefined framework conditions constraining their influence. The role of the social scientists, thus, is twofold. First, they must have a comprehensive knowledge of social science methodologies, they need to select the appropriate tools for the specific PM case, and they must be experts in applying these methods. Second, the social scientists facilitate the interaction between modelers and stakeholders both in terms of translating research questions into a form suitable for stakeholder discussions, and in terms of feeding back stakeholder comments and assessments to the modelers. Maintaining strict neutrality and concentrating on method and communication expertise are at the heart of the social scientists' facilitator role.

*Author contributions.*



| Dirk Scheer and Wilfried Konrad | Participatory modeling |
| Holger Class and Alexander Kissinger | Numerical modeling |
| Stefan Knopf and Vera Noack | Geological expertise and geological model setup |

*Acknowledgements.* The authors gratefully acknowledge the funding for the CO$_2$Brim project (03G0802A) provided by the German Federal Ministry of Education and Research (BMBF) and the German Research Foundation (DFG) within the geoscientific research and development program GEOTECHNOLOGIEN.



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
