# Peer review of "Regional-scale brine migration along vertical pathways due to CO2 injection – Part 1: the participatory modeling approach"

_Hydrology and Earth System Sciences, 2016_

## Referee Comment (RC1) · Anonymous Referee #1 · 8 Feb 2017

The manuscript addresses an important topic, i.e. the involvement of a wider community in modeling efforts. A novel approach is reported in the manuscript, i.e. inclusion of selected expert into the modelling process via interviews and a workshop. It is thus a novel approach and describes an experiment involving people. I do think this topic is relevant for this journal, and may help in the longer run to come to improved model scenarios and set-ups. The paper is clearly written, well structured and understandable. However, I do have suggestions and a number of issues with this paper, concerning the methods used as well as describing the experiment and the outcome more clearly, which I outline below. In the current version, no clear conclusions are formulated, just stated that this is a success. If it is a success, the authors should more clearly describe it. Just performing the interviews and the workshop is – to me – not enough, and clearer conclusions should be drawn. Also, the results should be better described, so that this work can become helpful for others. As stated above, the approach is novel and very interesting, so I would like to see this work reported.

The authors describe a so-called participatory approach to model setup in the context of brine migration driven potentially by carbon dioxide injection into a saline aquifer. The authors used two methods, i.e. eye-to-eye interviews and a workshop with discussion groups in varying composition to discuss and obtain opinions of the participants on certain model features and their importance.

This approach could contribute to a better model derivation, as the important features and effects to include in the model are discussed before and during the first modelling stages. I agree with the basic assumption that a wider participation would benefit here and may lead to answers that are more general from the modelling process. I thus think that this is a valuable research, which may in future help us solve geoscience-related questions in a more comprehensive way. However, I have suggestions to improve the manuscript quality. Firstly, this first part is basically a social-science approach, however published in a natural science journal. The manuscript would clearly benefit

- from a clearer description of the methods used. I am not familiar with these concepts

- a more open and wide literature review of approaches similar and used maybe in different fields. Discussions with stakeholders of certain topics are nothing new, they happen frequently i.e. between regulators, consultants and site owners. Also interesting here could be approaches used for finding disposal sites for hazardous wastes. It is difficult for me now to believe that this is the wider status of research on this field. Just think of the discussion groups in Germany initiated by the fracking discussion, I think there is something to learn here.

- a discussion of other possible methods not used. Why were they not used? Why did the authors use the interviews and the workshop, and why the "world café" format?

This would be a lot more informative, and help better understand the approach.

- Describe and discuss much clearer the choice of invited expert. At the end of the paper, there are a few sentences on this, but this of course is crucial. Inviting only natural scientists from regulators or science does not give the full spectrum of possible questions to be answered by the model. A wider participation could have shifted the model focus.

- a clear description of the questionnaire used and the questions asked. Why were they asked? What was the intention, and did that work out? Could the interviewees answer the questions asked by social scientists? Did they think them relevant? Generally, this questionnaire is probably very important, as it sets the whole scope. So why and how was this devised?

- How and why were the set of questions for the workshop devised? Why not other questions or other combinations of those?

I find the conclusions and results rather vague, very descriptive and repetitive. The manuscript does not allow a reproduction of the methods used, as they are not described. This requires more description of the results in the manuscript, and I suggest adding i.e. the questionnaire etc. in an appendix. In the current for, at least I could not transfer the approach used here to a similar topic, because not enough information and background is given. Especially concerning the background, a more general introduction into existing methods from social sciences would be helpful, as most Readers will be like me not familiar with the Terms used.

Also in the discussion section, a clear statement of the achievements would be helpful. Maybe this is also due to me being a natural scientist, but what are really the findings others could use? What are the individual lessons ? It just states that this was successful, but the success does not become very clear to me.

The abstract should be more concise and reflect the findings of the work.

---

## Referee Comment (RC2) · Anonymous Referee #2 · 13 Feb 2017

Review Comment to the manuscript "Regional-scale brine migration along vertical pathways due to CO 2 injection – Part 1: the participatory modeling approach" by Dirk Scheer et al.

I was involved already as a reviewer in the first round of reviewing. I find the applied changes of the manuscript in accordance to both reviewer comments suitable. I did numerous reviews in the past, however, this is the first time I review a social science paper. I believe it would be good if also a social scientist (in addition to us natural scientists) would have a look. To summarize I find the manuscript of relevance and worth to be published. I do have only some minor comments. In general there is (again) a tendency to use terms and phrases which are weakly defined. For instance,

page 2, line 13: it is not clear to me what is meant precisely with "producing and deploying conceptual and computer-based models". However, a these kind of phrases are common I do not insist on a change here. Page 2, Line 33: "in 2011", here the reference should be given. Page 2, Line 34: what is a Delphi survey? Page 3, Line 16: remove the word "short" here Figure 1: here also the backward iteration could be included Page 8, Line 9: before always 'brine' is referred to, now CO2 is named – is this consistent? Table 2: the text of the footnote should be put in the caption

---

## Author Comment (AC1) · 16 Mar 2017

by Dirk Scheer et al. dirk.scheer@kit.edu

Anonymous Referee #1

RC1-1: The manuscript addresses an important topic, i.e. the involvement of a wider community in modeling efforts. A novel approach is reported in the manuscript, i.e. inclusion of selected expert into the modelling process via interviews and a workshop. It is thus a novel approach and describes an experiment involving people. I do think this topic is relevant for this journal, and may help in the longer run to come to improved model scenarios and set-ups. The paper is clearly written, well structured and

understandable.

Answer: We appreciate very much the acknowledgment of our novel approach with integrating selected experts into the modeling process and the recognition this may yield into improved model scenarios and set-ups.

RC1-2: However, I do have suggestions and a number of issues with this paper, concerning the methods used as well as describing the experiment and the outcome more clearly, which I outline below. In the current version, no clear conclusions are formulated, just stated that this is a success. If it is a success, the authors should more clearly describe it. Just performing the interviews and the workshop is – to me – not enough, and clearer conclusions should be drawn. Also, the results should be better described, so that this work can become helpful for others. As stated above, the approach is novel and very interesting, so I would like to see this work reported. The authors describe a so-called participatory approach to model setup in the context of brine migration driven potentially by carbon dioxide injection into a saline aquifer. The authors used two methods, i.e. eye-to-eye interviews and a workshop with discussion groups in varying composition to discuss and obtain opinions of the participants on certain model features and their importance. This approach could contribute to a better model derivation, as the important features and effects to include in the model are discussed before and during the first modelling stages. I agree with the basic assumption that a wider participation would benefit here and may lead to answers that are more general from the modelling process. I thus think that this is a valuable research, which may in future help us solve geoscience-related questions in a more comprehensive way. However, I have suggestions to improve the manuscript quality. Firstly, this first part is basically a social-science approach, however published in a natural science journal.

Answer: Many thanks for summarizing and synthesising the major points of critique and recommendations. To our understanding the various issues tackled in the summarizing overview follow the bullet points below. Therefore, we will dedicate detailed responses

according to each bullet point subsequently.

The manuscript would clearly benefit

RC1-3: from a clearer description of the methods used. I am not familiar with these concepts

Answer: With re-reading our manuscript, we see the point and need to clearer describe the set of methods used in order to make it better understandable to non-social-scientists being the large majority of the HESS journal.

RC1-4: a more open and wide literature review of approaches similar and used maybe in different fields. Discussions with stakeholders of certain topics are nothing new, they happen frequently i.e. between regulators, consultants and site owners. Also interesting here could be approaches used for finding disposal sites for hazardous wastes. It is difficult for me now to believe that this is the wider status of research on this field. Just think of the discussion groups in Germany initiated by the fracking discussion, I think there is something to learn here.

Answer: We indeed have kept the literature in the submitted manuscript very short, and see the point to extend the literature review to better classify participatory approaches in the area of earth system and geological sciences. In case of the possibility to revise our manuscript, we will overwork and extend the literature reviews taking the mentioned recommendations into account.

RC1-5: a discussion of other possible methods not used. Why were they not used? Why did the authors use the interviews and the workshop, and why the "world café" format?

Answer: Within the Methods section, we will also include in a revised version a paragraph discussing methods used against other participatory approaches and deliver arguments for you selected approach. This would be a lot more informative, and help better understand the approach.

RC1-6: Describe and discuss much clearer the choice of invited expert. At the end of the paper, there are a few sentences on this, but this of course is crucial. Inviting only natural scientists from regulators or science does not give the full spectrum of possible questions to be answered by the model. A wider participation could have shifted the model focus.

Answer: This is a relevant point; we are grateful for this comment. We will give much more details within the methods section on expert selection and pros/cons on wider participation approaches.

RC1-7: a clear description of the questionnaire used and the questions asked. Why were they asked? What was the intention, and did that work out? Could the intervie-wees answer the questions asked by social scientists? Did they think them relevant? Generally, this questionnaire is probably very important, as it sets the whole scope. So why and how was this devised?

Answer: In the submitted version, we decided not to include too many details of the questionnaire in order to keep the manuscript short. However, we understand much more details on the questionnaire are necessary which will be dealt with in an revised version.

RC1-8: How and why were the set of questions for the workshop devised? Why not other questions or other combinations of those?

Answer: Thanks so much for making this clear. Within the method section, we will more explicitly describe how/why we set up the series of questions.

I find the conclusions and results rather vague, very descriptive and repetitive. The manuscript does not allow a reproduction of the methods used, as they are not de-scribed. This requires more description of the results in the manuscript, and I suggest adding i.e. the questionnaire etc. in an appendix. In the current for, at least I could not transfer the approach used here to a similar topic, because not enough information

and background is given. Especially concerning the background, a more general introduction into existing methods from social sciences would be helpful, as most Readers will be like me not familiar with the Terms used. Also in the discussion section, a clear statement of the achievements would be helpful. Maybe this is also due to me being a natural scientist, but what are really the findings others could use? What are the individual lessons ? It just states that this was successful, but the success does not become very clear to me. The abstract should be more concise and reflect the findings of the work.

Answer: to our understanding, this paragraph summarizes the work to be done for a revised version as laid out in the bullet points above more in details. In an revised version of the manuscript, we will focus on: (i) reworking the abstract to be more concise; (ii) the introduction section with an extended literature review on participatory approaches; (iii) the method section with more details on methods used (expert selection, questionnaire etc.); (iv) the results section with describing more specifically the details; (v) the discussion section with focussing on clear statements of achievements; (vi) and the conclusion section with explicitly highlighting the lessons learnt.

Anonymous Referee #2

RC2-1: I was involved already as a reviewer in the first round of reviewing. I find the applied changes of the manuscript in accordance to both reviewer comments suitable. I did numerous reviews in the past, however, this is the first time I review a social science paper. I believe it would be good if also a social scientist (in addition to us natural scientists) would have a look. To summarize I find the manuscript of relevance and worth to be published.

Answer: We are grateful the reviewer finds the applied changes suitable and the paper topic in general worth to be published.

RC2-2: I do have only some minor comments. In general there is (again) a tendency to use terms and phrases which are weakly defined. For instance, page 2, line 13: it

is not clear to me what is meant precisely with "producing and deploying conceptual and computer-based models". However, a these kind of phrases are common I do not insist on a change here.

Answer: In case of a revised version to be submitted, we will carefully got through the manuscript and improve to concisely define terms and phrases wherever it is suitable.

RC2-3: Page 2, Line 33: "in 2011", here the reference should be given.

Answer: We will provide a reference for this statement.

RC2-4: Page 2, Line 34: what is a Delphi survey?

Answer: Many thanks for hinting to the lack of details. We will include a short description on the method of a Delphi

RC2-5: Page 3, Line 16: remove the word "short" here

Answer: Many thanks for careful reading. We will delete the word "short".

RC2-6: Figure 1: here also the backward iteration could be included

Answer: This is true and would enhance better understanding by readers. We will do this.

RC2-7: Page 8, Line 9: before always 'brine' is referred to, now $CO_2$ is named – is this consistent?

Answer: We will check consistency of brine vs. $CO_2$.

RC2-8: Table 2: the text of the footnote should be put in the caption

Answer: We will put the footnote in the caption.

---

## Author Response (AR2)

**Author's Response**

**Commenting on Reviewer's comments to the paper: "Regional-scale brine migration along vertical pathways due to CO2 injection – Part 1: the participatory modeling approach"**

by Dirk Scheer et al.

Tel: ++49-(0)721-608-22994; Email: dirk.scheer@kit.edu

**Anonymous Referee #1**

**RC1-1:** The manuscript addresses an important topic, i.e. the involvement of a wider community in modeling efforts. A novel approach is reported in the manuscript, i.e. inclusion of selected expert into the modelling process via interviews and a workshop. It is thus a novel approach and describes an experiment involving people. I do think this topic is relevant for this journal, and may help in the longer run to come to improved model scenarios and set-ups. The paper is clearly written, well structured and understandable.

**Answer:** We appreciate very much the acknowledgment of our novel approach with integrating selected experts into the modeling process and the recognition this may yield into improved model scenarios and set-ups.

**RC1-2:** However, I do have suggestions and a number of issues with this paper, concerning the methods used as well as describing the experiment and the outcome more clearly, which I outline below. In the current version, no clear conclusions are formulated, just stated that this is a success. If it is a success, the authors should more clearly describe it. Just performing the interviews and the workshop is – to me – not enough, and clearer conclusions should be drawn. Also, the results should be better described, so that this work can become helpful for others. As stated above, the approach is novel and very interesting, so I would like to see this work reported. The authors describe a so-called participatory approach to model setup in the context of brine migration driven potentially by carbon dioxide injection into a saline aquifer. The authors used two methods, i.e. eye-to-eye interviews and a workshop with discussion groups in varying composition to discuss and obtain opinions of the participants on certain model features and their importance. This approach could contribute to a better model derivation, as the important features and effects to include in the model are discussed before and during the first modelling stages. I agree with the basic assumption that a wider participation would benefit here and may lead to answers that are more general from the modelling process. I thus think that this is a valuable research,

which may in future help us solve geoscience-related questions in a more comprehensive way. However, I have suggestions to improve the manuscript quality. Firstly, this first part is basically a social-science approach, however published in a natural science journal.

**Answer:** Many thanks for summarizing and synthesising the major points of critique and recommendations. To our understanding the various issues tackled in the summarizing overview follow the bullet points below. Therefore, we will dedicate detailed responses according to each bullet point subsequently.

The manuscript would clearly benefit

- **RC1-3:** from a clearer description of the methods used. I am not familiar with these concepts

    **Answer:** With re-reading our manuscript, we see the point and need to clearer describe the set of methods used in order to make it better understandable to non-social-scientists being the large majority of the HESS journal. In the revised version, we added in section 2.2 a paragraph on the explorative and qualitative design of the study that stipulates the choice of the methods and extended descriptions on the methods used

- **RC1-4:** a more open and wide literature review of approaches similar and used maybe in different fields. Discussions with stakeholders of certain topics are nothing new, they happen frequently i.e. between regulators, consultants and site owners. Also interesting here could be approaches used for finding disposal sites for hazardous wastes. It is difficult for me now to believe that this is the wider status of research on this field. Just think of the discussion groups in Germany initiated by the fracking discussion, I think there is something to learn here.

    **Answer:** We indeed have kept the literature in the submitted manuscript very short, and see the point to extend the literature review to better classify participatory approaches in the area of earth system and geological sciences. In the revised version, we added a paragraph on "involvement literature" in general contrasting the participatory from the generic involvement approach.

- **RC1-5:** a discussion of other possible methods not used. Why were they not used? Why did the authors use the interviews and the workshop, and why the "world café" format?

    **Answer:** Within the Methods section 2.2, we included reasons on method selection (explorative, qualitative).

This would be a lot more informative, and help better understand the approach.

- **RC1-6:** Describe and discuss much clearer the choice of invited expert. At the end of the paper, there are a few sentences on this, but this of course is crucial. Inviting only natural scientists from regulators or science does not give the full spectrum of possible questions to be answered by the model. A wider participation could have shifted the model focus.

**Answer:** This is a relevant point; we are grateful for this comment. In the revised paper, we elaborated more widely on recruitment criteria and choice of experts.

- **RC1-7:** a clear description of the questionnaire used and the questions asked. Why were they asked? What was the intention, and did that work out? Could the interviewees answer the questions asked by social scientists? Did they think them relevant? Generally, this questionnaire is probably very important, as it sets the whole scope. So why and how was this devised?

**Answer:** In the submitted version, we decided not to include too many details of the questionnaire in order to keep the manuscript short. However, we understand much more details on the questionnaire are necessary. We now added the interview guideline in the manuscript (Table 1).

- **RC1-8:** How and why were the set of questions for the workshop devised? Why not other questions or other combinations of those?

**Answer:** Thanks so much for making this clear. Within the results section, we now outlined elaboration of the set of questions as a close cooperation between modelers and social scientists against the background of discussing the two fundamental main assumptions of the modelling (space, realistic but not site-specific geo-model).

I find the conclusions and results rather vague, very descriptive and repetitive. The manuscript does not allow a reproduction of the methods used, as they are not described. This requires more description of the results in the manuscript, and I suggest adding i.e. the questionnaire etc. in an appendix. In the current for, at least I could not transfer the approach used here to a similar topic, because not enough information and background is given. Especially concerning the background, a more general introduction into existing methods from social sciences would be helpful, as most Readers will be like me not familiar with the Terms used. Also in the discussion section, a clear statement of the achievements would be helpful. Maybe this is also due to me being a natural scientist, but what are really the findings others could use? What are the individual lessons ? It just states that this was successful, but the success does not become very clear to me. The abstract should be more concise and reflect the findings of the work.

**Answer:** to our understanding, this paragraph summarizes the work to be done for a revised version as laid out in the bullet points above more in details. In the revised version of the manuscript, we carefully went through all sections and improved the paper on: (i) reworking the abstract to be more concise; (ii) the introduction section with an extended literature review on involvement literature; (iii) the method section with more details on methods used (expert selection, questionnaire etc.); (iv) the results section with describing more specifically the details; (v) the discussion section with focussing on clear statements of achievements; (vi) and the conclusion section with explicitly highlighting the lessons learnt.

35

**Anonymous Referee #2**

**RC2-1:** I was involved already as a reviewer in the first round of reviewing. I find the applied changes of the manuscript in accordance to both reviewer comments suitable. I did numerous reviews in the past, however, this is the first time I review a social science paper. I believe it would be good if also a social scientist (in addition to us natural scientists) would have a look. To summarize I find the manuscript of relevance and worth to be published.

**Answer:** We are grateful the reviewer finds the applied changes suitable and the paper topic in general worth to be published.

**RC2-2:** I do have only some minor comments. In general there is (again) a tendency to use terms and phrases which are weakly defined. For instance, page 2, line 13: it is not clear to me what is meant precisely with "producing and deploying conceptual and computer-based models". However, a these kind of phrases are common I do not insist on a change here.

**Answer:** In the revised version, we carefully went through the manuscript and improved several parts to concisely define terms and phrases wherever it was suitable.

**RC2-3:** Page 2, Line 33: "in 2011", here the reference should be given.

**Answer:** We replaced the reference within the sentence.

**RC2-4:** Page 2, Line 34: what is a Delphi survey?

**Answer:** Many thanks for hinting to the lack of details. We included a short description on the method of a Delphi.

**RC2-5:** Page 3, Line 16: remove the word "short" here

**Answer:** Many thanks for careful reading. We deleted the word "short".

**RC2-6:** Figure 1: here also the backward iteration could be included

**Answer:** This is true and would enhance better understanding by readers. We deleted the old Figure and elaborated a new one including backward iteration.

**RC2-7:** Page 8, Line 9: before always 'brine' is referred to, now $CO_2$ is named – is this consistent?

**Answer:** We checked the consistency of brine vs. $CO_2$ – as a result: $CO_2$ in this sentence is consistent.

**RC2-8:** Table 2: the text of the footnote should be put in the caption

**Answer:** We have put the footnote in the caption.

[revised manuscript text omitted]